# Parent and Child Choice of Sugary Drinks Under Four Labelling Conditions

**DOI:** 10.3390/nu17111920

**Published:** 2025-06-03

**Authors:** Zenobia Talati, Thomas McAlpine, Katlyn Mackenzie, Gael Myers, Liyuwork M. Dana, Jessica Charlesworth, Moira O’Connor, Caroline Miller, Barbara A. Mullan, Helen G. Dixon

**Affiliations:** 1School of Population Health, Curtin University, Bentley, WA 6102, Australia; thomas.mcalpine@curtin.edu.au (T.M.); kat.mackenzie@curtin.edu.au (K.M.); liyuwork.dana@curtin.edu.au (L.M.D.); jessica.charlesworth@curtin.edu.au (J.C.); m.oconnor@curtin.edu.au (M.O.); barbara.mullan@curtin.edu.au (B.A.M.); 2The Kids Research Institute Australia, Nedlands, WA 6009, Australia; 3Medical School, University of Western Australia, Crawley, WA 6009, Australia; 4enAble Institute, Curtin University, Bentley, WA 6102, Australia; 5Cancer Council Western Australia, Subiaco, WA 6008, Australia; gael.myers@cancerwa.asn.au; 6South Australian Health and Medical Research Institute, Adelaide, SA 5001, Australia; caroline.miller@sahmri.com; 7School of Public Health, University of Adelaide, SA 5001, Australia; 8Centre for Behavioural Research in Cancer, Cancer Council Victoria, Melbourne, VIC 3002, Australia; helen.dixon@cancervic.org.au; 9Melbourne School of Psychological Sciences, The University of Melbourne, Parkville, VIC 3052, Australia

**Keywords:** sugary drinks, sugar-sweetened beverages, child, parent, choice

## Abstract

**Background:** The majority of Australian children exceed the World Health Organization’s recommended dietary intake of free sugar, particularly through the consumption of sugar-sweetened beverages. Front-of-pack nutrition labels increase perceived risk and deter the consumption of sugar-sweetened beverages. However, past studies of young children have focused almost exclusively on a parent’s choice of beverage for children. This study investigated the influence of four label designs (text-based warning, tooth decay pictorial, teaspoons of sugar, and Health Star Rating) on the beverage choices of N = 1229 Australian children (aged 4–11 years) and their parents. **Methods:** In an online vending machine scenario, parent–child dyads were separately asked to select which beverage they would choose for themselves before and after being randomised to one label condition. The beverages displayed included 100% fruit juice, soft drink, soft drink with a non-nutritive sweetener, flavoured milk, plain milk and bottled water. Beverage healthiness was determined by a 1–10 rating based on a review by a panel of experts (10 dietitians and nutritionists). **Results:** Mixed-model ANOVAs showed that for parents, each label design performed comparably; however, for children, small but significant differences were seen in the effectiveness of different label designs, with the teaspoons of sugar label, text-based warning, and tooth decay pictorial found to be more impactful in promoting healthier drink choices than the Health Star Rating. **Conclusions:** These findings can inform public health advocacy efforts to improve food labelling and could be incorporated into educational resources to help children understand the nutritional profiles of different sugary drinks.

## 1. Introduction

### 1.1. Parent and Child Choice of Sugary Drinks Under Four Labelling Conditions

The World Health Organization (WHO) recommends that adults’ and children’s free sugar intake (that is, sugar added to foods and found naturally in honey, syrups and fruit juices) not exceed 10% of their total energy intake [1]. Excess sugar intake is a significant but modifiable risk factor for weight gain, cardiometabolic risks, dental caries and type 2 diabetes worldwide [2,3]. It is estimated that more than half of Australian children (60–70% depending on age) exceed the WHO recommendation [4]. Sugar-sweetened beverages (SSBs), defined by the WHO as all types of beverages containing free sugars [5], account for a significant proportion (37%) of sugar in the diet of Australian children [6]. Meta-analyses and systematic reviews have directly linked sugar-sweetened beverage intake with childhood obesity [7,8].

Drinks such as 100% fruit juice and flavoured milk tend to be perceived by parents as healthier than other SSBs like soft drinks despite their comparable sugar content [9,10,11,12]. This is in part because these drinks frequently display health nutrition and other marketing claims on the front label that highlight the presence of positive nutrients (e.g., vitamin C), ingredients (e.g., real fruit) or other beneficial attributes (e.g., “natural”) or the absence of unfavourable attributes (e.g., preservative-free), which position them as healthy or “better for you” options despite containing high amounts of sugar [12,13]. Such claims can bestow a “health halo” over unhealthy, child-oriented food products in parents’ eyes, [14] and it is likely that this cognitive bias also contributes to parents perceiving that SSBs displaying such claims are healthy options [13,15,16]. Given that some SSBs (especially fruit drinks and juice) are more likely to be targeted at and consumed in greater quantities by children [15,17], it is important that unbiased, easy-to-understand information is displayed on the front-of-pack of beverages to help consumers identify and select healthy options should they wish. Front-of-pack nutrition labels hold promise as a tool to provide consumers with accurate information on the sugar and energy content of drinks and potentially promote healthier drink choices and reduce daily energy intake [18,19].

The Health Star Rating (HSR) is a voluntary front-of-pack labelling initiative implemented in Australia and New Zealand. It was introduced by the governments of both countries in 2014 to help consumers make healthier food choices at a glance [20]. Few studies have assessed the effectiveness of the Australian HSR in relation to drinks [18,21,22,23], and the limited research, which has focused on adolescent and adult perceptions and drink selection behaviours, has produced mixed findings in terms of its effectiveness in reducing the consumption of unhealthy drinks [18,21,22,23]. This highlights the need to further explore the effectiveness of this label format, especially in comparison to other label formats that may be more effective, such as warning labels.

Following the successful implementation of tobacco warning labels worldwide, there have been calls to apply similar labels to SSBs due to their widespread over-consumption and associated health risks [24]. Meta-analyses of studies largely comprising adult participants show that health warning labels (such as teaspoons of sugar, text-based warnings and graphic health warnings) curb preferences and purchasing intentions for sugary drinks [25,26]. Warning labels depicting total sugars as teaspoon amounts have proven effective in influencing adults’ attitudes and intentions to consume SSBs due to their relatability, acceptability and simplicity [27,28,29].

Similarly, text-based warning labels describing the health risk, caloric content or sugar content of a drink have decreased adults’ SSB purchases and intentions to consume in a naturalistic randomised control trial [30]. Mathematical modelling estimates that in Mexico (where text-based warning labels have been implemented), applying these labels to non-alcoholic beverages could reduce obesity prevalence among adults by 14.7% in 5 years [31]. One study found that parents were significantly more likely to rate energy-dense nutrient-poor foods as less healthy when text-based warning labels were displayed on the package as compared to traffic light or dietary guideline labels [32], with similar results found in children aged 8–13 years [33] and adolescents aged 12–18 years [34]. Graphic health warning labels highlighting the health risks of sugar overconsumption have been found to reduce parents’ selection of SSBs for themselves and the intention to purchase these drinks for their children compared to control, caloric content only [28] and text-based [35] warning labels. Additionally, labels with an octagon shape (like a stop sign) are effective in drawing attention to energy-dense nutrient-poor foods compared to other label types [36,37,38,39].

While there have been studies on the effectiveness of these labels among adults and adolescents, few studies have assessed the effects of warning labels on beverage choices among young children. Across two recent meta-analyses [25,26] of studies on sugary drink warning labels, only one study [32] recruited children younger than 8. This is an important area for exploration given that young children are in a critical period for developing long-term eating and drinking habits [40]. As parents are important gatekeepers for their children’s diets [11], and children exert influence on the foods and drinks that parents purchase and provide for them, it is important to understand how various front-of-pack labelling schemes for SSBs impact both parents’ and their young children’s appraisal of and choice of drink options [41,42]. These insights can inform the design and implementation of public health interventions directed at informing and supporting healthier drink choices from an early age.

### 1.2. Aims and Hypotheses

To address the pervasive health risks associated with sugary drink consumption, front-of-pack labels are needed to counter the pervasive marketing of SSBs that parents and children are exposed to across a range of media and settings, especially at the point of sale. Building upon existing research underscoring the importance of accurate point-of-sale nutrition information and regulatory measures for mitigating the adverse health effects of regularly consuming SSBs, this study aimed to provide insights into the impact of different label formats on the healthiness of parents’ and children’s drink choices using a simulated vending machine scenario.

The labels included in this study were the Health Star Rating, the teaspoons of sugar label, a text-based warning label (with the text “WARNING: Drinks high in sugar contribute to tooth decay”) and a pictorial of a tooth with a lightning bolt (indicating tooth decay) and the text “HIGH IN SUGAR”. To our knowledge, these labels have not been tested in a sample of young children. In light of previous research, we hypothesise that among both parents and children, (i) the inclusion of any label will promote healthier drink choices compared to the control condition [25,26]; (ii) warning labels (text-based warning, teaspoons of sugar and tooth decay pictorial) will promote healthier drink choices than HSR, as they will evoke more emotional responses among children and adults [26]; and (iii) pictorial warning labels (teaspoons of sugar and the tooth decay pictorial) will promote healthier drink choices than the text-based warning label [18,21], especially among children who are still learning to read and may be more responsive to pictorial labels than text-based warning labels.

## 2. Methods

### 2.1. Design

A pre–post between- and within-subjects online experiment with four labelling intervention conditions—Health Star Rating (HSR) [20], text-based warning, teaspoons of sugar label and [43] tooth decay pictorial [44] (see Figure 1)—was conducted with parent–child dyads. Pre-exposure to a labelling condition, all participants viewed an image of a drink vending machine containing various SSBs and healthier drink options with no labels displayed (Figure 2) and chose their preferred drink. Participants then viewed the vending machine image again, with random allocation and exposure to their assigned labelling condition, and again chose their preferred drink. Images of the vending machine tasks under each labelling condition can be found in the Appendix A.

### 2.2. Participants

Parents of children between 4 and 11 years of age (recruited in age groups of 4–7 and 8–11 for equal sampling distribution) and residing in Australia were recruited. Participants had to be willing to complete the online survey with their child aged between 4 and 11 years and be one of the primary food providers for their child.

### 2.3. Measures

*Sociodemographics.* Parents were asked their age, postcode (which was converted to quintiles using the Index of Relative Socioeconomic Advantage and Disadvantage) [45], their gender identity, their highest level of education, whether they are the main grocery buyer in the household or whether this was a shared responsibility, their child’s age and their child’s gender identity.

*Vending machine image task.* Seven beverage types were shown in the vending machine, each with two options: (1) 100% fruit juice (orange and apple flavours), (2) fruit drink (orange and apple flavours), (3) soft drink (cola and lemonade/squash flavours), (4) soft drinks flavoured with non-nutritive sweetener (NNS beverages; cola and lemonade/squash flavours), (5) flavoured milk (strawberry and chocolate flavours), (6) plain milk (bottled and carton) and (7) plain bottled water (see Figure 2). Participants were asked “Imagine you have come to a vending machine, and you are thinking about buying yourself a drink. Please choose which drink you would buy in this situation, by tapping on that drink once. If you would not buy any of these drinks, please select this option underneath the image”. Additional instructions were also included for participants: “Tip: For a clearer image, we suggest pinching the screen on your mobile/tablet device or using the zoom function on your computer/laptop”. To control for brand preferences, all beverages had coloured labels and a generic beverage name (e.g., cola).

*Label conditions.* Parent–child dyads were randomised to one of four label conditions (Figure 2). Participants could zoom in on each image to see a larger version of the product label. The Health Star Rating (HSR) label, which is used in Australia, assigns 0–5 stars to indicate the overall healthiness of a product, with more stars representing better nutritional value. A Microsoft Excel spreadsheet downloaded from the HSR website was used to determine the HSR for each drink (besides water, which receives an automatic 5 stars) based on beverage ingredients [46]. The text-based warning label consisted of plain text stating “Drinks high in sugar contribute to tooth decay”. The tooth decay pictorial showed a tooth with a lightning bolt and the text “HIGH IN SUGAR” to highlight the risk of sugar on dental health. While graphic and lifelike photographs of decayed teeth have been used in previous studies, they are viewed as less feasible and acceptable by the general public [28,47]. Thus, a simpler, pictorial graphic indicating high sugar content was chosen for real-world application. Finally, the teaspoons of sugar label featured an image with a teaspoon of sugar alongside the number of teaspoons of sugar in the drink [29]. This was calculated by taking the total sugars per 100 mL in grams, multiplying by the total size in mL of the drink, dividing by 4 (i.e., the number of grams found in a level teaspoon of sugar) and rounding to the nearest 0.5 teaspoons.

For the second vending machine drink choice task, HSRs were displayed on all drink products in that condition, whereas the remaining label conditions were only displayed on those drinks classified as SSBs. Plain milk beverages did not include any warning labels due to the naturally occurring sugar (lactose) in these drinks. Water and beverages sweetened with NNS did not display warning labels due to their absence of added sugars. All labels, except the HSR label, were octagonal in shape with the health risk text in white and the word “WARNING” highlighted in red text.

*Label recall and prompted recognition.* Following the second vending machine task, where participants had not been told in advance that labels would be present, participants were asked, “Do you recall seeing a label on the drink you chose from the vending machine on the previous screen?” with response options of Yes, No and Unsure. For those that answered Yes or Unsure, this was followed by a multi-choice question “Of the following labels, which one do you recall seeing?*”* with each label image presented, as well as an additional “I do not recall the label shown” option. All participants were then shown the actual label that they were presented with.

*Label perceptions.* The perception questions were adapted from Scully et al. (2020) [21]. The questions were slightly rephrased for children to accommodate age-related differences in language and comprehension ability.

*Beverage selection healthiness.* Beverage healthiness was quantified through a review by a panel of experts. Specifically, the list of beverage categories was reviewed by a group of dietitians (n = 3) and nutritionists (n = 7) who independently and anonymously rated the healthiness of each beverage type from 1 to 10 (see Appendix A). Experts were asked to briefly outline the factors used to make their decision, with the level of processing, whether the drink is considered to be a core food or discretionary item as defined by the Australian Dietary Guidelines [48], the presence of negative nutrients (e.g., sugar, saturated fat) and the presence of positive nutrients (e.g., fibre, vitamin C, calcium) being the most commonly cited. The mean ratings for each category of beverages were used in the analyses.

### 2.4. Procedure

Following ethics approval for the study from Curtin University’s Human Research Ethics Committee (HRE2021-0693), recruitment via PureProfile (a third-party web panel provider) took place between February and March 2024. After reading the information sheet and providing informed consent (on behalf of themselves and their child), participants answered three bot detection check and eligibility questions. If an appropriate response was not provided, the data from these respondents were excluded.

Questions were labelled as either “Parent Questions” (in green) or “Child Questions” (in purple) throughout the survey. Quotas were implemented to ensure an approximately even randomisation of participants across the four conditions (teaspoons of sugar, tooth decay pictorial, text-based warning, HSR) and the two age groups of the children (i.e., ages 4–7 years and 8–11 years).

Parents were asked sociodemographic questions and then completed the first vending machine task with no labels on the beverages (Figure 2). They were then randomly allocated to one of the four label conditions and presented with the second vending machine task with the label superimposed on the front of relevant drinks (see Appendix A for each experimental stimulus). For each vending machine image (i.e., control and label conditions) there was an option for participants to select, “I would not buy any of these drinks”. Following the second vending machine image task, parents were asked whether they recalled the label they were shown and were then asked to select the label they saw. Following this, questions assessing perceptions of the specific label were asked, with an image of the label they were shown placed at the top of the same page in case they could not recall the label. The parent then handed the survey over to their child to complete the same two vending machine tasks and label recall and perception questions (with appropriate levels of parent supervision suggested depending on the age of the child). At the end of the survey, the participants were debriefed, thanked and redirected to the PureProfile website for monetary compensation in line with PureProfile protocols.

### 2.5. Data Analysis

All data analyses were conducted in SPSS (v. 29). Descriptive statistics were used to report the prevalence of different beverage choices for parents and children. To assess whether some label designs were recalled more easily than others, chi-square tests of contingencies were used to determine the relationship between correct label recall and label condition for parents and children, respectively. The ratings for each perception item were compared between label conditions using multiple one-way between group ANOVAs, corrected for multiple comparisons. Finally, mixed-model ANOVAs were used to detect the main effect of time (overall exposure to labels), the interaction between time and label condition, and the simple effects of time within each label condition for both parents and children.

## 3. Results

### 3.1. Sample Characteristics

Data from 77 participants were excluded because they completed the survey a second time, after either failing a bot check question or not meeting eligibility criteria in their first attempt. This left 1229 parent–child dyads. A small number of dyads (n = 25) stopped after completing the first parent vending machine task. The data they provided was used in relevant analyses relating to parents. Table 1 shows the participant characteristics for the total sample and by condition. No significant differences in any of these characteristics were observed between conditions, confirming that randomisation was successfully achieved.

### 3.2. Label Perceptions and Recall

A total of 793 (64.5%) parents and 636 (52.8%) children recalled seeing a label. Of these respondents, fewer accurately recalled the label they were shown (382 parents (43.8%) and 565 children (73.1%)). Parents were less accurate at recalling their assigned label than children (χ^2^ (1, N = 1206) = 64.11, *p* < 0.001). Of those who recalled seeing a label or those that were unsure, the HSR was most often recalled correctly by parents (67.4%), whereas the text-based label and the teaspoons of sugar label were poorly recalled (28.7% and 29.1%, respectively). For children, the HSR was also correctly recalled the most (79.7%), whereas the text-based warning label was the most poorly recalled (67.9%). Furthermore, 334 (38.3%) parents and 117 (15.1%) children indicated that they were unable to recall which label they saw. See Table 1 for further details.

Table 2 shows parents’ and children’s perceptions of each label. Generally, parents and children tended to rate the HSR label less favourably than the other label conditions for every perception item, whereas the teaspoons of sugar and the tooth decay pictorial labels were perceived similarly favourably, followed by the text-based warning label. For younger children (age 4–7 years), there were no significant differences between the labels on each of the 11 perception items, except that children rated the text-based warning and the teaspoons of sugar label significantly higher than HSRs for making them feel worried. For older children (8–11 years), however, the tooth decay pictorial generally tended to be rated the highest across most perception items and was rated significantly higher than the HSR label for three of the perceptions: easy to understand, tells the truth and makes me feel worried. The text-based warning was also rated significantly higher than the HSR for some perception items (easy to understand, made me feel worried). In contrast to parents, however, the teaspoons of sugar label did not rate significantly higher than the HSR across any of the perception items for children 8–11 years old.

### 3.3. Beverage Choice by Product Type

Table 3 and Table 4 show parents’ and children’s choice of beverage product pre- and post-exposure to the labelling intervention. For parents, water, NNS beverages and soft drinks were the most chosen beverages when viewed without the warning label (see Table 3). Following exposure to label condition, these three drink categories remained the most chosen across all conditions. Table 4 shows the detailed percentages of children’s chosen beverages for each condition. Unlike for parents, differences in beverage selections between conditions were statistically significant for children (χ^2^ (21, N = 1203) = 36.13, *p* = 0.021). Flavoured milk, 100% fruit juice and soft drinks were the most chosen when drinks were viewed without a label. This pattern persisted following exposure to the HSR label; however, preferences were slightly different for other label conditions. In the text-based warning label condition, fruit drinks were the second most chosen beverage, in place of 100% fruit juice, while flavoured milk remained the most chosen and soft drinks were second. Under the tooth decay pictorial label condition, soft drinks lost favour while flavoured milk remained the most preferred, and water was chosen second most alongside 100% fruit juice in third. The teaspoons of sugar warning label showed a similar pattern, with flavoured milk remaining the most preferred, and 100% fruit juice and water being chosen second and third, respectively.

### 3.4. Beverage Choice by Healthiness Rating

The majority of participants (73.7% of both parents and children) did not change their choice following exposure to a label. A mixed-model ANOVA showed that for parents, the main effect of time on the healthiness of the beverage selected was significant (*F* (1, 1136) = 19.40, *p* < 0.001, partial η^2^ = 0.02), suggesting that overall, beverages selected after exposure to a label (*M* = 5.38, *SD* = 3.48) were significantly healthier than those selected before exposure (*M* = 5.04, *SD* = 3.38). However, the interaction between time and condition was not significant, indicating that the effect of time was generally consistent across conditions (*F* (3, 1136) = 0.97, *p* = 0.405, partial η^2^ = 0.00). Table 5 shows the change in healthiness rating scores for each condition. Notably, the HSR and tooth decay pictorial conditions showed significant improvements in healthiness ratings, whereas the text-based warning and teaspoons of sugar label conditions improved the healthiness in beverage choice, although not significantly.

Although there were no demographic differences across conditions, suggesting that randomisation was effective, associations between the healthiness of parents’ and children’s beverage selections and each demographic variable were tested to determine whether demographic factors played a role in selection preferences following label exposure. For parent’s beverage choices, the results showed that there was a significant main effect of parent’s gender whereby women selected healthier beverages than men across both time points (*F* (1, 1132) = 132.74, *p* = 0.010), although this effect was small (partial η^2^ = 0.01). For parents, there were no significant interactions between the parent’s gender and label condition, indicating that the effect of labels was consistent for both men and women. No other significant main or interaction effects were detected for any other demographic variable (parent age, education, SES, main grocery buyer status, child gender or child age).

For children, a mixed-model ANOVA showed that, like parents, there was a significant main effect of time on the healthiness of the beverage selected (*F* (1, 1172) = 4, *p* < 0.001, partial η^2^ = 0.04), suggesting that overall, beverages selected after exposure to labels (*M* = 4.78, *SD* = 2.66) were healthier than those selected before exposure (*M* = 4.39, *SD* = 2.40). The interaction between time and condition was also significant, indicating that this effect differed across conditions (*F* (3, 1172) = 5.20, *p* = 0.001, partial η^2^ = 0.01). Further examination of the simple effects of time within each condition (Table 5) showed that healthiness ratings significantly increased from pre- to post-exposure for all conditions except the HSR label.

The effects of parent and child demographics on children’s beverage choices were also tested. Parents’ age, gender, education level, SES and main grocery buyer status were not related to the healthiness of children’s beverage choices. However, there were very small but significant main effects for both the age (*F* (1, 1174) = 103.49, *p* = 0.002, partial η^2^ = 0.01) and gender of the child (*F* (1, 1171) = 49.94, *p* = 0.030, partial η^2^ = 0.00), whereby younger children and girls selected healthier beverages than their counterparts, although the effects were very minor. No interaction effects were detected, indicating that the effects of the labels on child beverage healthiness selection were consistent across demographic factors.

Given that not all participants accurately recalled the label to which they were exposed, sensitivity analyses were conducted for both parents and children. Among the subset of parents who accurately recalled the label they saw, the mixed-model ANOVA demonstrated similar patterns of results, whereby the main effect of time was significant (*F* (1, 364) = 13.96, *p* < 0.001, partial *η*^2^ = 0.04). However, the interaction was significant (*F* (3, 364) = 3.83, *p* = 0.010, partial *η*^2^ = 0.03), revealing that only parents in the HSR label and the teaspoons of sugar label conditions chose significantly healthier drinks following label exposure (see Table 5).

For the subset of children who correctly recalled the label they were exposed to, the main effect of time remained significant (*F* (1, 555) =29.95, *p* < 0.001, partial *η*^2^ = 0.05). However, the interaction between time and condition was not significant in this subset of children (*F* (3, 555) = 1.37, *p* = 0.250, partial *η*^2^ = 0.01,) indicating that the effect of time was consistent across conditions. However, an examination of the simple effects showed the same pattern of results in this subset as the full sample of children, whereby all conditions except the HSR label demonstrated significantly improved healthiness ratings (Table 5).

## 4. Discussion

This present research aimed to investigate the impact of different label formats on parent and child beverage choice using a simulated vending machine scenario. The results provided support for all hypotheses among both the children and parents in our sample. The findings contribute to the growing body of evidence on the effectiveness of warning labels in shaping consumer behaviour, particularly in the context of reducing SSB consumption among children [49]. While the overall impacts of the labels were modest in terms of changing selection, some specific formats of front-of-pack labels demonstrated more consistent effects on promoting healthier beverage choices. Children who saw the tooth decay pictorial and teaspoons of sugar labels selected soft drinks less and water more often. However, flavoured milk remained the most selected drink among children across all conditions, including those featuring warning labels. This may reflect a strong preference for these drinks that is not easily countered by front-of-pack labels.

For children, the results consistently demonstrated that exposure to three of the four label conditions led to the selection of healthier beverages. The text-based warning label showed the most modest change, followed by the tooth decay pictorial label, with the largest changes resulting from exposure to the teaspoons of sugar label. In contrast, the HSR label did not significantly impact the healthiness of children’s beverage selection. This may have occurred as a result of the warning labels making it easier for children to distinguish between unhealthy SSBs and healthier options because they only appeared on the unhealthy SSBs, whereas HSRs appeared on all drinks irrespective of healthiness (but with varying numbers of stars). This pattern was consistent regardless of whether children accurately recalled the label or not, although the effects were slightly less pronounced in the subset of children who did recall the label accurately. This has important implications for health policy in Australia, as it provides some evidence that HSR, the current labelling system, may have minimal influence on the healthiness of children’s beverage choices.

The novel labels had a significant but small effect on the healthiness of the beverages children selected, suggesting that these labels were effective in conveying the health risks associated with sugary drinks. There are several plausible possibilities for this effect, with the most obvious being that children may be more familiar with the HSR label, which was first implemented in 2014 [20]. Children are more likely to want to learn about atypical information [50], which may have prompted deeper processing of the novel labels in comparison to the HSR. Alternatively, the effect may be attributed to the concrete and relatable nature of the novel warnings (e.g., tooth decay is bad), which might resonate more with children’s understanding of health risks, whereas the HSR provides a more abstract representation of healthiness. This is supported by the findings from the perception ratings of each of the label conditions, which showed that the HSR label was rated the hardest to understand. Additionally, for children in the older age bracket (8–11 years), the HSR label elicited significantly less worry and was less likely to make them stop and think than other label conditions. This is in line with previous research showing that health communications that evoke stronger cognitive and emotional engagement, such as worry or reflective thinking, may be more persuasive in influencing behaviour, as both emotional arousal and thoughtful processing have been shown to enhance message effectiveness in public health contexts [51,52]. It is important to be aware that the changes observed, although significant, were weak, suggesting that brief experimental exposure to the labels did not have a marked impact on parents or children.

For parents, exposure to front-of-pack labels also resulted in significantly healthier beverage selections than when beverages did not include a label. This was true for parents who accurately recalled seeing the label and for the total parent sample. Interestingly, these effects were driven by different conditions in the total sample and the subset who recalled seeing the label. For the total sample, the tooth decay pictorial led to the greatest improvement in the healthiness of beverage choice, but in the subset that correctly recalled the label, the teaspoons of sugar label had the greatest effect size of any condition. This shows that when noticed, the teaspoons of sugar condition could be the most effective label for influencing beverage selection decisions. This is in line with previous studies demonstrating the effectiveness of this warning label design [27,28,29]. However, as the majority of parents did not correctly recall the label they had seen, policy makers should consider which warning label could be most effective in influencing beverage choice at a population level.

In contrast to children, the HSR label elicited significant and positive improvements in the healthiness of drink choices for both samples of parents (correct recall and total sample). This positive change was observed despite significantly lower ratings for almost every perception item compared to the other label conditions. This suggests that even though the text-based warning and the tooth decay pictorial were easier to understand, more believable, more convincing and more likely to make parents feel concerned than the HSR label, the effect on the healthiness of beverage selection were either comparable (tooth decay pictorial) or inferior (text-based warning label) to the HSR label.

The relatively low overall change, however, suggests that while some parents may be influenced by labelling, broader or more intensive interventions might be needed to significantly shift beverage purchasing and consumption patterns at the population level. This is an important consideration for public health interventions, as parental purchasing decisions directly influence the availability of beverages for their children [53,54].

The overall low recall rates of the labels, particularly among parents, is in line with previous research, including a study of 1200 adults across 12 countries that found an average recall rate of 62.2% across the sample and lower recall specifically for a text-based warning label (48.4%) and the HSR (56.5%) [55]. These findings highlight a crucial challenge in the effectiveness of front-of-pack labelling: if consumers do not notice or remember the labels, the potential impact on their choices is significantly diminished. To maximise the effectiveness of these labels, they could be designed to be more eye-catching, and they must be used in combination with complementary strategies such as consumer education, consistent messaging across various platforms and regulatory support that can help reinforce the importance of these labels, ultimately guiding consumers towards healthier choices [56,57]. Additionally, understanding the contexts in which labels are most often overlooked can provide insights into optimising label placement and design to ensure that they are more frequently noticed by consumers.

### Limitations and Future Directions

While this study provides valuable insights, some limitations should be acknowledged. First, the reliance on self-reported data for beverage choice introduces the potential for bias. Although the experiment was designed to simulate a vending machine and responses were anonymous, social desirability bias may have inflated the proportion of healthier choices. Given that participants were randomised to the conditions, we expect this effect to be equal across conditions and between-group inferences to be unaffected. Additionally, assessing drink choices immediately after a single, brief exposure to a labelling intervention limits the ability to draw causal conclusions about the impact of warning labels on long-term beverage choices. Future research could, therefore, assess how multiple exposures to the same label influence beverage selection, which may also help enhance label recall.

This study measured label effectiveness through changes in the healthiness of beverage choices from no label to label. Although this allowed for a more direct observation of the impact of various label conditions compared to no label, it is possible that exposure to the no label condition first influenced participants’ choices when the label was present. Future studies could randomise participants to start with either no label or label conditions to better understand the effect of label exposure. Finally, this study used a simulated vending machine scenario to explore the impact of labels on choice and may not generalise to other scenarios where beverages are purchased. Future research could examine the effectiveness of these labels in real-world settings such as schools, supermarkets or convenience stores to better understand their influence in more naturalistic environments.

## 5. Conclusions

This study highlights the differential impact of front-of-pack labels on beverage choices between parents and children, with children showing greater sensitivity to novel labels with pictorial elements, whereas parents were more influenced by the familiar HSR label and the teaspoons of sugar label. The findings underscore the importance of designing warning labels that are both memorable and persuasive, particularly for younger audiences. The teaspoons of sugar label could be the most optimal label in positively influencing beverage selection and improving diets, given that this was consistently effective for both parents and children; however, consideration must be given to increasing the salience of labels, especially for parents. By understanding how warning labels influence choices, interventions can be designed to promote healthier drink choices from an early age. To maximise the effectiveness of these labels, they should be integrated into broader public health strategies that address the various factors influencing beverage choices.

## Figures and Tables

**Figure 1 nutrients-17-01920-f001:**
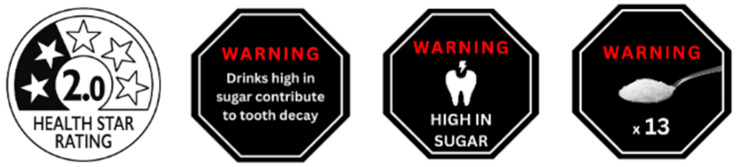
The experimental label conditions. Note: from left to right, the Health Star Rating label (adjusted to show the star rating of each drink) was displayed on all drinks in the vending machine, whereas the text-based warning label, the tooth decay pictorial label and the teaspoons of sugar label (adjusted to show the teaspoons of sugar in each drink) were only displayed on drinks with added sugar in the vending machine.

**Figure 2 nutrients-17-01920-f002:**
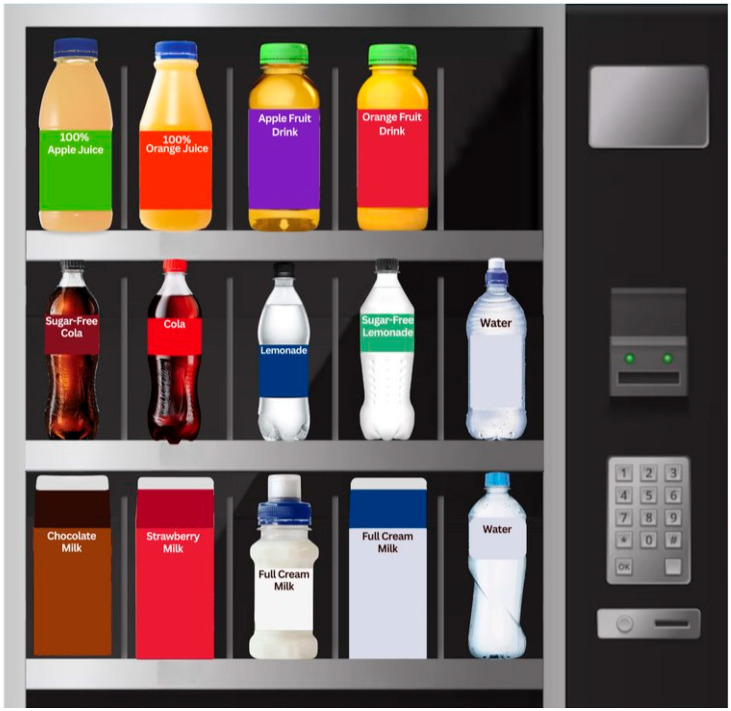
Vending machine task pre-label exposure.

**Table 1 nutrients-17-01920-t001:** Parent and child participants’ sociodemographic characteristics and label recall for the total sample and by label condition.

		Condition			
Variable	Total (N = 1229)	HSR (n = 305)	Text-Based (n = 301)	Tooth Pictorial (n = 310)	Teaspoons of Sugar (n = 298)
**Parents**					
**Age** (M (SD))					
Years	39.10 (7.65)	39.16 (7.74)	38.87 (7.68)	39.32 (7.71)	38.96 (7.61)
**Gender** (n (%))					
Male	430 (35.0)	101 (33.1)	108 (35.9)	108 (34.8)	109 (36.6)
Female	792 (64.4)	202 (66.2)	192 (63.8)	199 (64.2)	189 (63.4)
Non-binary	4 (0.3)	1 (0.3)	1 (0.3)	2 (0.6)	0 (0.0)
Prefer not to say	2 (0.2)	1 (0.3)	0 (0.0)	1 (0.3)	0 (0.0)
**Main Grocery Buyer** (n (%))					
Mainly me	1055 (85.8)	264 (86.6)	256 (85.0)	273 (88.1)	251(84.2)
Mainly someone else	23 (1.9)	5 (7.5)	7 (2.3)	2 (0.6)	9 (3.0)
Equally shared	150 (12.2)	36 (11.8)	38 (12.6)	35 (11.3)	38 (12.8)
**Highest Education Level** (n (%))					
Primary	5 (0.4)	2 (0.7)	0 (0.0)	1 (0.3)	2 (0.7)
Secondary	189 (15.4)	43 (14.1)	49 (16.3)	47 (15.2)	48 (16.1)
TAFE/Technical College	402 (32.7)	111 (36.4)	98 (32.6)	94 (30.3)	95 (31.9)
University, undergraduate	406 (33.0)	94 (30.8)	90 (29.9)	112 (36.1)	106 (35.6)
University, postgraduate	226 (18.4)	55 (18.0)	64 (21.2)	56 (18.1)	47 (15.8)
**SES Quintile** (n (%))					
1	146 (11.8)	37 (12.1)	42 (14.0)	29 (9.4)	37 (12.4)
2	195 (15.8)	50 (16.4)	49 (16.3)	45 (14.5)	48 (16.1)
3	252 (20.5)	68 (22.3)	59 (19.6)	59 (19.0)	62 (20.8)
4	272 (22.1)	71 (23.3)	63 (20.9)	73 (23.5)	61 (20.5)
5	363 (29.6)	79 (25.9)	88 (29.2)	104 (33.5)	90 (30.2)
**Label Recall** (n (%))					
Yes	793 (65.8)	211 (70.3)	197 (66.1)	213 (68.7)	172 (57.9)
No	80 (6.6)	22 (7.3)	20 (6.7)	12 (3.9)	26 (8.8)
Cannot Say	332 (27.6)	67 (22.3)	81 (27.2)	85 (27.4)	99 (33.3)
Correct Recall	382 (48.2)	157 (67.4)	62 (28.7)	105 (46.7)	58 (29.1)
**Child**					
**Age** (M (SD))					
Years	7.43 (2.31)	7.45 (2.31)	7.45 (2.30)	7.46 (2.26)	7.45 (2.30)
**Age-group** (n (%))					
4–7 yrs old	602 (50)	150 (50)	149 (50)	155 (50)	148 (50)
8–11 yrs old	600 (50)	149 (50)	148 (50)	154 (50)	149 (50)
**Gender** (n (%))					
Male	666 (54.2)	162 (53.1)	162 (53.8)	164 (52.9)	171 (57.4)
Female	558 (45.4)	141 (46.2)	138 (45.8)	144 (46.5)	127 (42.6)
Non-binary	1 (0.1)	0 (0.0)	1 (0.3)	0 (0.0)	0 (0.0)
Prefer not to say	4 (0.3)	2 (0.6)	0 (0.0)	2 (0.6)	0 (0.0)
**Label Recall** (n (%))					
Yes	636 (52.8)	148 (49.5)	159 (53.5)	164 (52.9)	165 (55.4)
No	137 (35.8)	39 (13.0)	31 (10.4)	30 (9.7)	37 (12.4)
Cannot Say	431 (11.4)	112 (37.5)	107 (36.0)	116 (37.4)	96 (32.2)
Correct Recall	565 (73.1)	149 (79.7)	129 (67.9)	145 (74.7)	142 (70.6)

Note: Group sizes may not add up to the total number, as a small portion of participants partially completed the survey. Full information is therefore reported where possible.

**Table 2 nutrients-17-01920-t002:** Mean perceptions of each label by condition.

		Condition							
		HSR		Text-Based		Tooth Pictorial		Teaspoons of Sugar	
**Perception**	η^2^	M	SD	M	SD	M	SD	M	SD
**Parent**		(n= 298)		(n = 296)		(n = 309)		(n = 297)	
Easy to understand *^,✝^	0.03	5.55	1.27	6.08 ^a^	1.02	6.06 ^a^	1.19	5.90 ^a^	1.30
Believable *^,✝^	0.06	5.20	1.41	5.96 ^a^	1.08	5.97 ^a^	1.19	5.75 ^a^	1.18
Relevant to me *^,✝^	0.02	5.24	1.41	5.51	1.31	5.71 ^a^	1.18	5.58	1.20
Made me stop and think *^,✝^	0.03	4.70	1.69	5.08	1.57	5.19 ^a^	1.61	5.50 ^a^	1.39
Would talk to others about *	0.04	4.16	1.73	4.67 ^a^	1.70	4.81 ^a^	1.63	5.13 ^a^	1.54
Taught me something new *^,✝^	0.06	4.18	1.74	3.72	1.88	4.22	1.84	**4.97**	1.53
Convincing *^,✝^	0.03	4.90	1.56	5.32 ^a^	1.33	5.43 ^a^	1.36	5.54 ^a^	1.22
Made strong argument *^,✝^	0.03	4.90	1.62	5.43 ^a^	1.39	5.26	1.61	5.62 ^a^	1.35
Made me feel concerned *	0.07	4.25	1.56	4.94 ^a^	1.59	5.05 ^a^	1.55	5.45 ^ab^	1.39
Effective *^,✝^	0.04	4.91	1.58	5.23	1.45	5.41 ^a^	1.44	5.67 ^a^	1.21
Exaggerated	0.01	3.66	1.58	3.37	1.69	3.65	1.72	3.37	1.59
**Children (4–7 yrs old)**		**(n = 150)**		**(n= 149)**		**(n= 155)**		**(n = 148)**	
Easy to understand	0.02	4.11	1.86	4.62	1.62	4.75	1.54	4.39	1.78
Tells the truth	0.01	4.79	1.45	5.17	1.16	5.00	1.34	5.02	1.28
Matters to me	0.01	4.06	1.74	4.44	1.50	4.27	1.59	4.26	1.76
Made me stop and think	0.01	3.69	1.90	4.29	1.69	4.14	1.69	4.08	1.76
Would talk to others about	0.01	3.88	1.86	4.32	1.69	4.24	1.74	4.26	1.79
Taught me something new	0.01	4.40	1.80	4.70	1.49	4.49	1.49	4.88	1.56
Made me change my mind	0.01	3.44	1.80	3.83	1.68	3.66	1.79	3.88	1.90
Made strong argument	0.02	4.31	1.82	4.99	1.45	4.55	1.63	4.70	1.61
Made me feel worried *	0.05	3.48	1.67	4.52 ^a^	1.52	4.12	1.65	4.20 ^a^	1.77
Effective	0.01	4.57	1.69	4.95	1.45	4.80	1.52	4.86	1.54
Exaggerated	0.01	3.46	1.63	3.74	1.41	3.70	1.46	3.94	1.52
**Children (8–11 yrs old)**		**(n = 149)**		**(n = 148)**		**(n = 154)**		**(n = 149)**	
Easy to understand *^,✝^	0.08	4.79	1.57	5.57 ^a^	1.22	5.79 ^a^	0.99	5.21	1.43
Tells the truth *	0.04	4.91	1.23	5.37	1.23	5.51 ^a^	1.14	5.08	1.37
Matters to me	0.02	4.24	1.57	4.83	1.58	4.79	1.52	4.48	1.57
Made me stop and think *	0.03	4.14	1.71	4.75	1.58	4.81	1.61	4.67	1.59
Would talk to others about	0.01	3.77	1.77	4.09	1.79	4.25	1.64	4.09	1.73
Taught me something new	0.02	4.69	1.56	4.59	1.59	4.80	1.60	5.21	1.40
Made me change my mind	0.01	3.84	1.74	4.30	1.84	4.28	1.88	4.26	1.82
Understand why I should drink less	0.02	4.44	1.63	5.09	1.45	4.95	1.64	4.72	1.62
Made me feel worried *	0.03	3.72	1.63	4.45 ^a^	1.66	4.44 ^a^	1.73	4.32	1.71
Effective	0.01	4.90	1.28	5.06	1.39	5.25	1.39	5.03	1.50
Exaggerated	0.01	3.70	1.59	3.72	1.58	3.98	1.65	3.79	1.54

Note: * Indicates a significant (*p* < 0.001) difference between condition means as determined by a one-way between-group ANOVA. ^✝^ Due to violations in normality and/or homogeneity of variance, mean differences here were also tested using a Kruskal–Wallis test and no difference in the pattern or significance in findings was observed. η^2^ = effect size. **Bold face** indicates a condition mean that was significantly (*p* < 0.001) higher than all three other means. ^a^ indicates a significantly higher (*p* < 0.001) mean than the HSR condition. ^b^ indicates a significantly higher (*p* < 0.001) mean than the text-based warning condition.

**Table 3 nutrients-17-01920-t003:** Parent’s beverage selection by frequency (%) pre- and post-exposure to label conditions.

	Beverage Type	Pre-Label Exposure (N = 1229)	Condition			
			HSR (n = 305)	Text-Based (n = 301)	Tooth Pictorial (n = 311)	Teaspoons of Sugar (n = 298)
Parents	Water	322 (26.2)	88 (28.9)	95 (31.6)	105 (33.8)	96 (32.1)
	NNS Beverage	258 (21.0)	70 (23.0)	75 (24.9)	72 (23.2)	67 (22.4)
	Soft Drink	237 (19.3)	54 (17.7)	43 (14.3)	51 (16.4)	47 (15.7)
	100% Juice	167 (13.6)	40 (13.1)	38 (12.6)	32 (10.3)	27 (9.0)
	Fruit Drink	80 (6.5)	18 (5.9)	16 (5.3)	11 (3.5)	20 (6.7)
	Flavoured Milk	70 (5.7)	18 (5.9)	17 (5.6)	15 (4.8)	18 (6.0)
	No Purchase	63 (5.1)	7 (2.3)	7 (2.3)	19 (6.1)	18 (6.0)
	Full Cream Plain Milk	32 (2.6)	10 (3.3)	10 (3.3)	6 (1.9)	5 (1.7)

**Table 4 nutrients-17-01920-t004:** Children’s beverage selection by frequency (%).

	Beverage Type	Pre-Label Exposure (N = 1205)	Condition			
			HSR (n = 299)	Text-Based (n = 297)	Tooth Pictorial (n = 310)	Teaspoons of Sugar (n = 298)
Children	Flavoured Milk	316 (26.2)	77 (25.8)	80 (26.9)	57 (18.4)	64 (21.5)
	100% Juice	241 (20.0)	52 (17.4)	36 (12.1)	49 (15.8)	58 (19.4)
	Soft Drink	212 (17.6)	49 (16.4)	40 (13.4)	35 (11.3)	41 (13.8)
	Fruit Drink	196 (16.3)	46 (15.4)	51 (17.2)	40 (12.9)	34 (11.4)
	Water	107 (8.9)	26 (8.7)	33 (11.1)	55 (17.7)	44 (14.8)
	NNS Beverage	82 (6.8)	30 (10.0)	33 (11.1)	39 (12.6)	35 (11.7)
	Full Cream Plain Milk	32 (2.7)	13 (4.3)	20 (6.7)	25 (8.1)	17 (5.7)
	No Purchase	19 (1.6)	6 (2.0)	4 (1.3)	10 (3.2)	5 (1.7)

**Table 5 nutrients-17-01920-t005:** Effect of front-of-pack label condition on healthiness scores of beverage choice for all respondents and for only those who correctly recalled the label.

		All respondents			Respondents Who Correctly Recalled the Label		
	Condition	Mean Change	SE	95% CI [LL, UL]	Mean Change	SE	95% CI [LL, UL]
Parents	HSR	0.37 *	0.15	[0.07, 0.67]	0.48 *	0.21	[0.08, 0.89]
	Text-based	0.25	0.15	[−0.05, 0.55]	−0.21	0.33	[−0.85, 0.44]
	Tooth Pictorial	0.54 ***	0.15	[0.24, 0.83]	0.49	0.25	[−0.01, 0.99]
	Teaspoons of Sugar	0.20	0.16	[−0.11, 0.50]	1.38 ***	0.34	[0.72,2.04]
Children	HSR	−0.01	0.12	[−0.25, 0.23]	0.22	0.17	[−0.12, 0.56]
	Text-based	0.43 ***	0.12	[0.20, 0.67]	0.47 *	0.19	[0.10, 0.83]
	Tooth Pictorial	0.53 ***	0.12	[0.29, 0.76]	0.55 **	0.18	[0.20, 0.89]
	Teaspoons of Sugar	0.62 ***	0.12	[0.38, 0.89]	0.71 ***	0.18	[0.37, 1.06]

Note: A negative coefficient represents a decrease in the healthiness rating of the beverages selected. SE = Standard Error. UL = Upper Limit. LL = Lower Limit. * *p* < 0.05 at the Bonferroni-corrected level. ** *p* < 0.01 at the Bonferroni-corrected level. *** *p* < 0.001 at the Bonferroni-corrected level.

## Data Availability

The data presented in this study are available on request from the corresponding author. The data are not publicly available due to ethical restrictions.

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
