# Peer review of "Parent and Child Choice of Sugary Drinks Under Four Labelling Conditions"

_nutrients, 2025, doi:10.3390/nu17111920_

Round 1

Reviewer 1 Report

Comments and Suggestions for Authors

This manuscript reports the investigation of the influence of four label designs (text-based warning, tooth decay pictorial, teaspoons of sugar, Health Star Rating) on the beverage choices of N = 1,229 Australian children (aged 4-11 years) and their parents. 

Were the images presented large enough for the participant to see the message? What are the dimensions of the vending machine image on a cell phone and a desk top computer screen? Could the participant adjust the size of the image on the screen to enhance the visibility of the label? Was any attention drawn to the label in moving from the no-label to label conditions?

Instead of using a subjective professional rating of healthiness, why not use as the dependent variable a more objective number of teaspoons of sugar selected?

Do we have any idea of how often and among whom the parent and child discussed the answers to the questions, and how that influenced the responses?

Table 1 should present the values for each of the 4 groups and the group specific recall data.

Please identify the sample size at the top of each column in Tables 1 and 2.

In the Discussion the authors state that “The novel labels had a significant effect on the healthiness of the beverages children selected, suggesting these labels were effective in conveying the health risks associated with sugary drinks”. However, in the results the authors report most parents and children did not change the drink they selected. This suggests to me that the labels were not generally effective. The Discussion needs to be revised to emphasize the very weak effects obtained. Obtaining weak effects is important to report too. Either other labels need to be developed and tested or parents and children are not responsive to labels. – As the authors noted, many factors likely influence the selection of a beverage, and these labels had a minimum effect. Can analyses be conducted to assess the effects of the demographic characteristics on the lack of change in beverage selection?

The labels were rather anodyne. Future research should involve graphic artists to generate labels that are more likely to draw attention and influence memory and effect.

Reviewer 2 Report

Comments and Suggestions for Authors

Dear Authors,

The manuscript presents the results of an interesting experiment that has relevance to the effectiveness of shaping health-promoting purchasing behavior through different beverage labeling systems. The strength of the presented study is the involvement of parent-child pairs as participants. This made it possible to detect differences between parents and children regarding the impact of different labels on their choices.

The manuscript is well structured, the methodology, including the experiment, was written in detail and clearly. The discussion of the results is interesting. However, I suggest expanding it a bit to include the issue of labeling as a tool for reducing dietary energy supply from SSBs. Only once, in the Introduction section (on the second page) did the authors note that “Meta-analyses and systematic reviews have directly linked sugar-sweetened beverage intake with childhood obesity.7,8”. The entire manuscript, on the other hand, is dominated by the issue of reducing the risk of tooth decay, especially as two types of labeling (the text-based warning label, and the tooth decay pictorial label) relate to it. Therefore, I suggest raising the possibility of overweight/obesity risk reduction in the context of a label showing the number of teaspoons of sugar in a beverage. After all, the results show that this label proved to be the most influential in beverage choice decisions.

And four more minor suggestions:

P. 2.

…” beverages (SSBs), defined by the WHO as all types of beverages containing free sugars,5 account for a significant proportion of sugar in the diet of Australian children”.

- readers are curious about the facts, so if the 2011-2012 Australian Health Survey identifies this significant proportion, please provide how much.

P. 4.

It seems to me that the numbering of Figures 1 and 2 in the “Design” text was confused. If so, the numbering should also be corrected in the rest of the manuscript (e.g., page 5).

P. 5.

Chapter “Design” - it can be added at the end that photos of the vending machine with drinks labeled with particular symbols are included as supplementary materials.

Chapter “Participant” - it should be added whether parents were asked to give their and their child's informed consent to participate in this study.

Kind regards
